# Improved Structural Discovery and Representation Learning of Multi-Agent Data

## Abstract

Central to all machine learning algorithms is data representation. For multi-agent systems, selecting a representation which adequately captures the interactions among agents is challenging due to the latent group structure which tends to vary depending on context. However, in multi-agent systems with strong group structure, we can simultaneously learn this structure and map a set of agents to a consistently ordered representation for further learning. In this paper, we present a dynamic alignment method which provides a robust ordering of structured multi-agent data enabling representation learning to occur in a fraction of the time of previous methods. We demonstrate the value of this approach using a large amount of soccer tracking data from a professional league.

## 1 Introduction

The natural representation for many sources of unstructured data is intuitive to us as humans: for images, a 2D pixel representation; for speech, a spectrogram or linear filter-bank features; and for text, letters and characters. All of these possess fixed, rigid structure in space, time, or sequential ordering which are immediately amenable for further learning. For other unstructured data sources such as point clouds, semantic graphs, and multi-agent trajectories, such an initial ordered structure does not naturally exist. These data sources are set or graph-like in nature and therefore the natural representation is unordered, posing a significant challenge for many machine-learning techniques.

A domain where this is particularly pronounced is in the fine-grained multi-agent player motions of team sport. Access to player tracking data changed how we understand and analyze sport (Miller et al., 2014; Franks et al., 2015; Wei et al., 2013; Cervone et al., 2014; Power et al., 2017; Sha et al., 2016; 2018; Yue et al., 2014). More relevantly, sport has risen to an increasingly key space within the machine learning community as an application to expand our understanding of adversarial multi-agent motion, interaction, and representation (Lucey et al., 2013; Le et al., 2017; Felsen et al., 2018; Zheng et al., 2016; Zhan et al., 2018; Yeh et al., 2019; Kurach et al., 2019).

In sport there exists strong, complex group-structure which is less prevalent in other multi-agent systems such as pedestrian tracking. Specifically, the *formation* of a team captures not only the global shape and structure the group, but also enables the ordering of each agent according to a "role" within the group structure. In this regard, sport possesses relational structure similar to that of faces and bodies, which can be represented as a graph of key-points. In those domains, representation based on a fixed key-point ordering has allowed for cutting edge work across numerous tasks with a variety of approaches and architectures (Antonakos et al., 2015; Cootes et al., 2001; Akhter et al., 2012; Joo et al., 2015; Simon et al., 2017).

Unlike for faces and bodies, the representation graph in sport is dynamic as players constantly move and switch positions. Thus dynamically discovering the appropriate representation of individual players according to their role in a formation affords us structural information while learning a useful representation for subsequent tasks. This challenge was addressed by the original role-based alignment of Lucey et al. (2013) and subsequently by Bialkowski et al. (2016) and Sha et al. (2017). Role-based alignment allows us to take unstructured multi-agent data, and reformat it into a consistent vector format that enables subsequent machine learning (Fig. 1).

Here we formulate the role-based alignment as consisting of the phases of formation discovery and role assignment. Formation discovery uses unaligned data to learn an optimal *formation template*; during role assignment a bipartite mapping is applied between agents and roles in each frame to

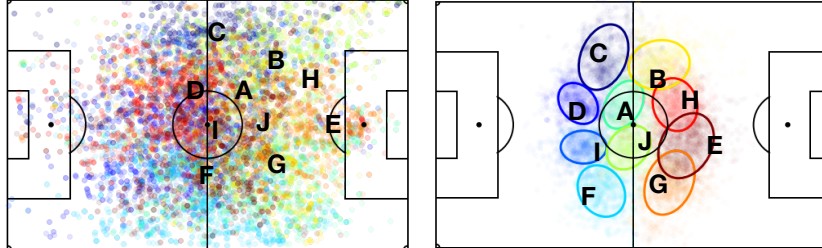

Figure 1: A structure representation enables machine learning of multi-agent data. (Left) Data-points are colored according to the agent identity (letters denote agents in a given frame). (Right) By learning and aligning data to a formation template, we represent agents in a consistent vector form conducive to learning. Agents are now ordered by the role to which they are assigned.

produce "aligned data". A major limitation in past approaches was the speed of the template discovery process. In this work we propose an improved approach to the above alignment methods which provides faster and more optimal template discovery for role-based representation learning. Importantly, we seek to learn the same representation as Lucey et al. (2013), Bialkowski et al. (2016), and Sha et al. (2017) by maximizing the same objective function of Bialkowski et al. (2016) in a more effective manner. The reduced computational load enables on-the-fly discovery of the formation templates, new context-specific analysis, and rapid representation learning useful for modeling multi-agent spatiotemporal data.

Our main contributions are: the formulation of this problem as a three-step approach (formation discovery, role assignment, template-clustering), the use of soft-assignment in the formation discovery phase thereby eliminating the costly hard-assignment step of the Hungarian Algorithm (Kuhn, 1955), a resetting training procedure based on the formation eigenvalues to prevent spurious optima, quantification of the impact of initialization convergence and stability, a restriction of the training data to key-frames for faster training with minimal impact on the learned representation, and a multi-agent clustering framework which captures the covariances across agents during the template-clustering phase.

## 2 BACKGROUND

### 2.1 REPRESENTING STRUCTURED MULTI-AGENT DATA

A collection of agents is by nature a set and therefore no defined ordering exists *a priori*. To impose an arbitrary ordering introduces significant entropy into the system through the possible permutations of agents in the imposed representation.

To circumvent this, some approaches in representing multi-agent tracking data in sport have included sorting the players based on an "anchor" agent (Mehrasa et al., 2018). This is limiting in that the optimal anchor is task-specific, making the representation less generalizable. An "image-based" representation (Yue et al., 2014; Zheng et al., 2016; Miller et al., 2014) eliminates the need for an ordering, however, this representation is lossy, sparse, and high-dimensional.

The role-based alignment protocol for sport of Lucey et al. (2013) used a codebook of hand-crafted formation templates against which frame-level[1] samples were aligned. This work was extended by Bialkowski et al. (2016) which learned the template directly from the data. Sha et al. (2017) further employed a hierarchical template learning framework, useful in both retrieval (Sha et al., 2018) and trajectory prediction (Felsen et al., 2018; Yeh et al., 2019). Le et al. (2017) similarly learned an agent-ordering directly from the data by learning separate role-assignment and motion-prediction policies in an iterative and alternating fashion.

### 2.2 PERMUTATION-EQUIVARIANT APPROACHES

Permutation-equivariant approaches seek to leverage network architectures which are insensitive to the ordering of the input data. Approaches using graph neural networks (GNN) (Kipf & Welling, 2016; Gilmer et al., 2017; Battaglia et al., 2018; 2016; Hoshen, 2017) have become very popular and shown tremendous promise. These approaches are particularly valuable for tasks (e.g. pedestrian tracking) which lack the strong coherent group structure of sport and therefore cannot leverage meth-

---

[1]Throughout we use the term "frame" to indicate a single moment in time in reference to the data being obtained via optical tracking from video.

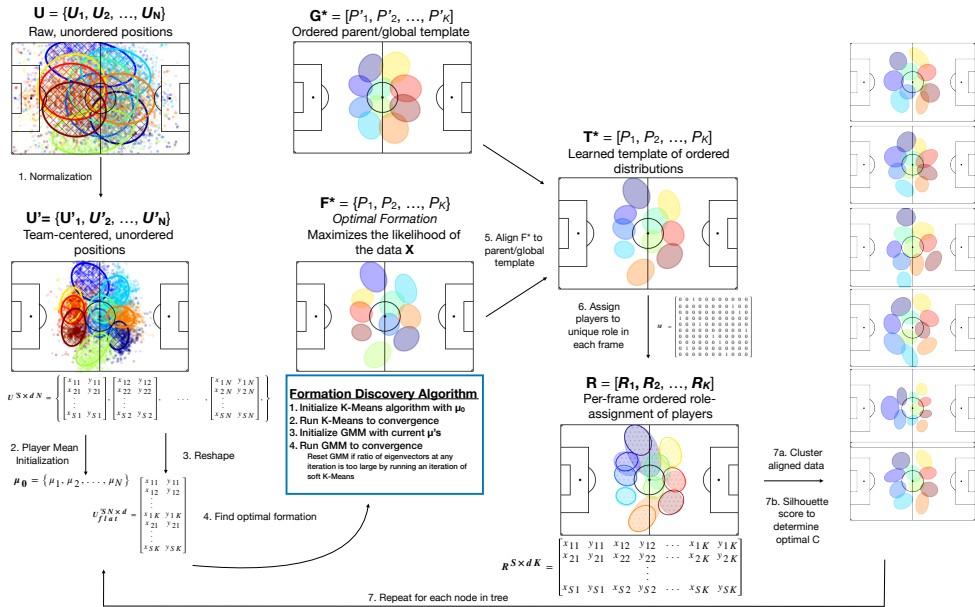

Figure 2: An overview of the proposed method. The procedure consists of (1) Normalization, (2) Initialization, (3) Reshaping, (4) Formation Discovery, (5) Template Alignment, (6) Role Assignment, (7) Template Clustering. In the role assignment step, the template distributions are shown as unfilled thick ellipses and the observed distributions of the role-aligned data are shown as the textured ellipses.

ods such as role-alignment. Within sport, Kipf et al. (2018) used a GNN to predict the trajectories of players while simultaneously learning the edge-weights of the graph. Yeh et al. (2019) demonstrated the advantages in using GNNs to forecast the future motion of players in sports, surpassing both the role (Bialkowski et al., 2016) and tree-based approaches (Sha et al., 2017) on most metrics.

The success of these approaches, however, does not negate the value of role-based alignment. The learned formation structure provides valuable insight into high-level organization of the group. Furthermore, many traditional machine learning techniques and common deep architectures require an ordered-agent representation. This is again similar to what is seen in the modeling of faces and bodies: great success has been achieved using geometric deep learning (Monti et al., 2017; Kipf et al., 2018), but approaches based on a fixed representation remain popular and effective (Taylor et al., 2017; Kanazawa et al., 2019; Arnab et al., 2019; Walker et al., 2017; Rayat Imtiaz Hossain & Little, 2018).

Interesting work has also been done to learn permutations for self-supervised feature learning or ranking tasks (Adams & Zemel, 2011; Mena et al., 2018; Cruz et al., 2017). Central to these approaches is the process of Sinkhorn normalization (Sinkhorn & Knopp, 1967), which allows for soft-assignment during the training process and therefore a flow of gradients. Exploring the application of Sinkhorn normalization to this task is beyond the scope of this current work, however, we provide additional context on this method in Section A.2.

# 3 APPROACH

## 3.1 PROBLEM FORMULATION

Mathematically, the goal of the role-alignment procedure is to find the transformation $A : \{\boldsymbol{U}_1, \boldsymbol{U}_2, \ldots, \boldsymbol{U}_N\} \times \boldsymbol{M} \mapsto [\boldsymbol{R}_1, \boldsymbol{R}_2, \ldots, \boldsymbol{R}_K]$ which maps the unstructured set $\boldsymbol{U}$ of $N$ player trajectories to an ordered set (i.e. vector) of $K$ role-trajectories $\boldsymbol{R}$. [2] Each player trajectory is itself an ordered set of positions $\boldsymbol{U}_n = [x_{s,n}]_{s=1}^S$ for an agent $n \in [1, N]$ and a frame $s \in [1, S]$. We recognize $\boldsymbol{M}$ as the optimal permutation matrix which enables such an ordering.

Thus our goal is to find the most probable set $\mathcal{F}^*$ of 2D probability density functions where

$$\mathcal{F}^* = \arg\max_{\mathcal{F}} P(\mathcal{F}|\boldsymbol{R}) \tag{1}$$

---

[2]Generically $N$ need not equal $K$ as a player may be sent off during a game, but for simplicity it is safe to assume $N = K$ in this work.

$$P(\boldsymbol{x}) = \sum_{n=1}^{N} P(\boldsymbol{x}|n)P(n) = \frac{1}{N} \sum_{n=1}^{N} P_n(\boldsymbol{x}). \tag{2}$$

Bialkowski et al. (2016) transforms this equation into one of entropy minimization where the goal is to minimize the amount of overlap (i.e. the KL-Divergence) between each role. The final optimization equation in terms of the total entropy $H$ then becomes

$$\boldsymbol{\mathcal{F}}^* = \arg\min_{\boldsymbol{\mathcal{F}}} \sum_{n=1}^{N} H(\boldsymbol{x}|n). \tag{3}$$

See A.1 for additional details.

The authors then use expectation maximization (EM) to approximate this solution and note similarity to k-means clustering. However, as they represent that data non-parametrically in terms of per-role heat maps, hard assignment must be applied at each iteration so the distributions may be updated. Instead, we note that equation 2 describes the occupancy of space by any agent in any point in time as a mixture of conditional distributions across each of the $N$-roles. This is further equivalent to the sum over $n$-generating distributions. Thus if we model these generating distributions as $d$-dimensional Gaussian distributions, this reduces the template-discovery process to that of a Gaussian Mixture Model.

## 3.2 TOY PROBLEM FORMULATION

Understanding the notion of independence under the different formulations of this problem is key. This may be better understood by considering a toy problem: imagine we have three independent 1D Gaussian distributions we wish to sample $S$ times from each. It is known that we sample from each distribution in rounds, effectively generating the samples in "triplets", although the order within the triplets is random. We then seek to reassign the points back to their original distributions.

Following the approach of Bialkowski et al. (2016), the "structure" imposed by the triplet sampling is enforced through the hard-assignment at each iteration. Recall, however, that the original distributions were statistically independent; the triplet structure we wish to respect is imposed by the assignment step, not the underlying distributions.

Contrastingly, in our method the samples are treated as fully independent; had all samples been taken from the first distribution, followed by the second, followed by the third, the outcome at the "distribution-discovery" phase would be identical to that having sampled the data in rounds. Only after the three distributions are estimated would the assignment of each point in every triplet be assigned to the distribution which maximized the overall likelihood in that triplet.

Besides being more computationally effective (see Section 4.2), this allows us to find the true MLE of the distributions. Our method will always discover a more optimal estimate of Eq. 2. This can be understood in considering how the assignment is performed during optimization. For each triplet, the likelihood of assigning each point to each distribution is computed in both approaches. In our approach, this gives *the* likelihood under each mixture component. In the hard-assignment approach, however, if two (or more) points in a triplet have their highest likelihood under the same component, the exclusionary assignment *must* result in a lower likelihood than assigning each point to its preferred Gaussian.

Furthermore, in our approach, each sample contributes to every component of the mixture, thus the data under the mixture remains "fixed" during the optimization process. In contrast, as the hard-assignments are made, the samples contributing to each distribution changes each iteration. This, in combination with the sub-optimal likelihood above, effectively "breaks" the expectation maximization step and can cause solutions to diverge or oscillate, which is inconsistent with a maximum likelihood solution which must monotonically increase.

Thus our approach is computationally efficient, more intuitively captures the independence of the generating distributions versus the structure of the sampling, and ensures a likelihood function that will converge under expectation maximization.

## 3.3 FORMATION-DISCOVERY

Our procedure explained here is presented visually in Figure 2 and algorithmically in Algorithm A.3.

Data is normalized so all teams are attacking from left to right and have mean zero in each frame, thereby removing translational effects. Following the approach of Bialkowski et al. (2016), we initialize the cluster centers for formation-discovery with the average player positions. The impact of this choice of initialization is explored in Section 4.5.

We now structure all the data as a single $(SN) \times d$ vector where $S$ is the total number of frames, $N$ is the total number of agents (10 outfielders in the case of soccer), and $d$ is the dimensionality of the data (2 here). The K-Means algorithm is initialized with the player means calculated above and run to convergence; we find that running K-Means to convergence produces better results than running a fixed number of iterations as is commonly done for initialization. The cluster-centers of the last iteration are then used to initialize the subsequent mixture of Gaussians.

Mixture of Gaussians are known to suffer from component collapse and becoming trapped in pathological solutions. To combat this, we monitor the eigenvalues $(\lambda_i)$ of each of the components throughout the EM process. If the eigenvalue ratio of any component becomes too large or too small, the next iteration runs a Soft K-Means (i.e. a mixture of Gaussians with spherical covariance) update instead of the full-covariance update. We find that the range $\frac{1}{2} < \frac{\lambda_1}{\lambda_2} < 2$ works well. In practice, we find this is often unnecessary when analyzing a single game as the player-initialization provides the necessary stabilization, but becomes important for analysis over many teams/games where that initialization signal is weaker. We refer to this set of $K$ distributions which maximizes the likelihood of the data the *Formation*, which we denote $\mathcal{F}^*$.

Note that the formation is a set of distributions. To enforce an ordering, we must align to a parent template, $\boldsymbol{G}^*$, which is an ordered set of distributions. The specific ordering of this template is unimportant so long as it is established and fixed. We align $\mathcal{F}^*$ to $\boldsymbol{G}^*$ by finding the Bhattacharyya distance (Bhattacharyya, 1943) between each distribution in $\mathcal{F}^*$ and $\boldsymbol{G}^*$ given by

$D_B = \frac{1}{8}(\mu_{\mathcal{F}^*{}_i} - \mu_{\boldsymbol{G}^*_j})^T \sigma^{-1}(\mu_{\mathcal{F}^*{}_i} - \mu_{\boldsymbol{G}^*_j}) + \frac{1}{2}\ln(\frac{\det \sigma}{\sqrt{\det \sigma_{\mathcal{F}^*{}_i} \det \sigma_{\boldsymbol{G}^*_j}}})$ where $\sigma = \frac{\sigma_{\mathcal{F}^*{}_i} + \sigma_{\boldsymbol{G}^*_j}}{2}$ to

create a $K \times K$ cost matrix and then use the Hungarian algorithm to find the best assignment. We have now produced our *Template*, $\mathcal{T}^*$ an ordered set of distributions with an established ordering that maximizes the likelihood of the data.

## 3.4 ROLE-ASSIGNMENT

The process of role-assignment maps each player in each frame to a specific role with the restriction that only one player may occupy a role in a given frame. We find the likelihood that each agent belongs to each of the discovered distributions in each frame which was already calculated during the formation-discovery step. This produces a $N \times K$ cost matrix in each frame; the Hungarian algorithm is again used to make the optimal assignment. Thus we have achieved the tasks of formation-discovery and role-assignment having had to apply the Hungarian algorithm on only a single pass of the data. We now represent the aligned data as a $S \times (dK)$ matrix $\boldsymbol{R}$.

## 3.5 CLUSTERING MULTI-AGENT DATA

With an established well-ordered representation, we are now able to cluster the multi-agent data to discover sub-templates and perform other analysis. Sub-templates may be found either through flat or hierarchical clustering. Generically, we seek to find a set of clusters $\boldsymbol{C}$ which partitions the data into distinct states according to:

$$\arg\min_{\boldsymbol{C}} \sum_{C_k \in \boldsymbol{C}} \sum_{\boldsymbol{R}_i, \boldsymbol{R}_j \in C_k} \|P(\boldsymbol{R}_i) - P(\boldsymbol{R}_j)\|_2 \tag{4}$$

For flat clustering, a $dN$-dimensional K-Means model is fit to the data. To help initialize this clustering, we seed the model with the template means plus a small amount of noise. To determine the optimal number of clusters we use a measure similar to Silhouette score (Rousseeuw, 1987):

$$\mathbb{E}(\boldsymbol{R}) = \frac{1}{|\boldsymbol{R}|} \sum_{C_k \in \boldsymbol{C}} \sum_{\boldsymbol{R}_i \in C_k} \frac{\|P(\boldsymbol{R}_i) - \mu_{kn}\|_2 - \|P(\boldsymbol{R}_i) - \mu_k\|_2}{\|P(\boldsymbol{R}_i) - \mu_{kn}\|_2} \tag{5}$$

where $\mu_k$ is the mean of the cluster that example $\boldsymbol{R}_i$ belongs to and $\mu_{kn}$ is the mean of the closest neighbor cluster of example $\boldsymbol{R}_i$. Equation 5 measures the dissimilarity between neighboring clusters

and the compactness of the data within each cluster. By maximizing E we seek to capture the most discriminative clusters.

To learn a tree of templates through hierarchical clustering, we follow the method of (Sha et al., 2017) with minor modification on how the clusters and templates are initialized. Algorithm A.4 outlines this procedure.

# 4 RESULTS

## 4.1 DATASET

For this work, we used an entire season of player tracking data from a professional European soccer league, consisting of 380 games, 6 of which were omitted due to missing data. The data is collected from an in-venue optical tracking system which records the $(x, y)$ positions of the players at 10Hz. The data also contains single-frame event-labels (e.g. pass, shot, cross) in associated frames; these events were used only to identify which frames contained the onset of an event which we call *event-frames*. Unless explicitly noted, the analysis used only event-frames for training, providing over 1.8million samples across the season.

## 4.2 RUN COMPLEXITY

Finding the optimal solution to K-Means is NP-hard, even for 2 clusters. However, through standard methods K-means clustering can achieve an average per-iteration complexity of $(samples \cdot clusters \cdot dimensions)$ while Gaussian mixture models have a complexity of $(samples \cdot clusters \cdot dimensions^2)$ per iteration due to the additional calculation of the precision matrix (Lloyd, 1982; Verbeek et al., 2003). Note that for all algorithms, $samples$ becomes $SN$ since each agent in each frame contributes to the distributions. The Hungarian Algorithm has a complexity of $elements^3$ per application.

In the original algorithm of Bialkowski et al. (2016), the cost matrix per frame is calculated in a manner resembling that of the GMM, requiring the full distribution (i.e. mean and precision matrix) to be computed so the likelihoods may be calculated. However, the Hungarian Algorithm is then applied across the $N$-agents in each of the $S$-frames. This produces a per-iteration complexity of $(SN)Kd^2N^3$. With $K = N$, this simplifies to $SN^5d^2$ (see Table 1 of Section A.5). In contrast, K-Means and GMM have a per-iteration complexity of $SN^2d$ and $SN^2d^2$, respectively. Therefore, for a sport like soccer, $N = 10$, causing the hard-assignment based algorithm to be $\sim 1000$ times slower than the proposed approach.

## 4.3 COMPARISON OF DISCOVERED $\mathcal{F}^*$

As all of these methods are unsupervised, there is no notion of a "more accurate" formation. However, as the goal is to find the $\mathcal{F}^*$ which maximizes the likelihood of the data, for each team-game-period (TGP), we computed the formation via hard-assignment and our current method, and computed the per-sample average log-likelihood of the data under each method. Our method produced a lower (i.e. more likely) log-likelihood for *every* TGP-formation, consistent with the theoretical guarantees (Figure 3A). In general, the difference between the two approaches was very small, an average difference in log-likelihood of 0.028.

We compute the field area covered by a role as $A = \frac{\pi}{\sqrt{\lambda_1 \lambda_2}}$ where $\lambda_1$ and $\lambda_2$ are the eigenvalues of the covariance matrix for that role. On average, the field area covered by a role under the current method is $0.021m^2$ smaller than the corresponding role under Bialkowski et al. (2016). This is consistent with the formations learned via the two methods being extremely similar; the average KL-Divergence (Kullback & Leibler, 1951) between corresponding roles under the two method is 0.14nats.

## 4.4 COMPRESSION EVALUATION

Template-based alignment has been shown to produce a compressed representation of multi-agent spatiotemporal data (Lucey et al., 2013). We repeat this analysis here in Figure 3. Similar to the approach in (Sha et al., 2017), we evaluate the compressibility of the approach using clustering and principle component analysis (PCA). We randomly selected 500,000 frames from the larger data set. Frames were aligned according to Algorithm A.3 ("Role_current" in Figure 3), via hard-assignment

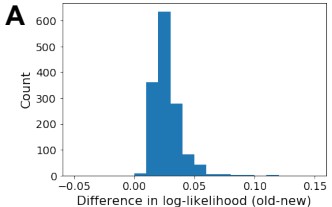 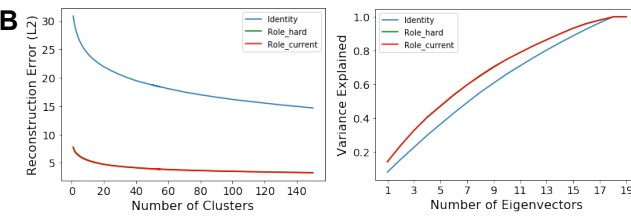

Figure 3: (A) Difference in the per sample log-likelihood under the template learned calculated via hard-assignment and the current method. All of the values are positive demonstrating the formation learned under the current method always captures the data better. (B) Template-based alignment produces a more compressed representation of the data than an identity-based representation. Left: reconstruction error as a function of the number of clusters. Right: variance accounted for as a function of the number of eigenvectors used. In both *Role_current* corresponds to the method of the current work *Role_hard* (often directly under Role_current) corresponds the hard-assignment approach.

("Role_hard"), or left unordered ("Identity"). K-means clustering was applied to both the original unaligned data and aligned data for varying values of K. The average within-cluster-error (WCE) was calculated according to $\text{WCE} = \frac{1}{|\boldsymbol{R}|} \sum_{C_k} \sum_{\boldsymbol{R}_i \in C_k} \|\boldsymbol{R}_i - \mu\|_2$ where we again abuse $C_k$ to indicate the $k^{th}$ cluster after K-means clustering. Similarly we run PCA on both the unaligned and aligned data and compute the variance explained by the eigenvectors: $\text{Variance Explained} = \frac{\lambda_k}{\sum_{i=1}^{D} \lambda_i}$ where $\lambda_i$ is the $i^{th}$ eigenvalue indicating the significance of the $i^{th}$ eigenvector.

Role-based representation, regardless of the method used to compute it, is significantly more compressive than an identity-based representation. The representation computed via the current method is slightly more compressive than under hard assignment: the per player reconstruction error over the range in Figure 3B is on average 0.76m lower and the variance explained on average is 0.068 higher.

## 4.5 IMPACT OF INITIALIZATION AND KEY-FRAMES

The original template-learning procedure proposed initializing the algorithm with the distributions of each player, as players tend to spend much of their time in a specific role. In the subsequent work of Sha et al. (2017), a random initialization at each layer was proposed.

To assess the impact of the player-mean initialization, we ran Alg. A.3 20 times per-TGP, each of which contains about 1500 frames, and recorded the reconstruction error during the K-Means initialization phase of the algorithm. While the exact reconstruction error is sample specific, all samples showed the same trend as Figure 4A: player-mean initialization begins with a much lower reconstruction error and converges significantly more quickly, often within 10 steps. In contrast, the random initialization is much more variable, takes many more iterations to converge, and often does not converge to as good a solution.

The use of event-only "key frames" is also a key performance and stabilization measure. Limiting the data to event-only frames reduces the data by a factor of $\sim 10$, producing a speed-up of the same factor. This has minimal impact on the learned template as seen in Figure 4B. In most instances, the templates learned are almost identical: the average L2-distance between the center of two role distributions is 0.24m and the average Bhattacharyya distance (Bhattacharyya, 1943) between two role distributions is 0.078.

## 4.6 CONTEXT-SPECIFIC FORMATIONS

Previously, due to the slowness of the hard-assignment approach, templates had to be learned as a part of a preprocessing step before storage/analysis/consumption. Usually this would be done at the TGP-level, generating a total of 4 "specialist" templates per game. In contrast, the proposed method allows templates to be computed "on the fly". For several thousand rows of data, the formation can be discovered and aligned in only a few seconds. This allows us to select data under interesting contexts and learn the template that best describes those scenarios across many games.

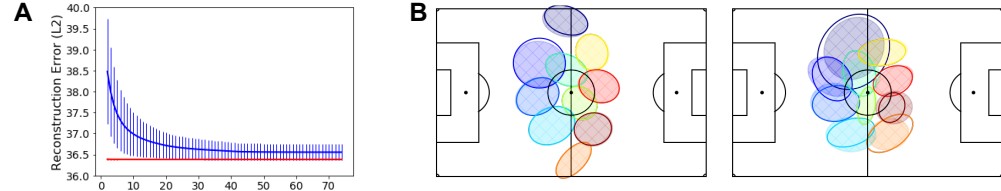

Figure 4: Impact of initialization and key-frame selection. (A) Player-mean initialization (red) enables the K-Means initialization to run to convergence in fewer iterations than random (blue) initialization. (B) Learning the formation on event-only "key-frames" (thick line, no hashing) results in formations which are very similar to the formations learned on all data (thin line, hashing), but runs significantly faster due the reduced data size and is less prone to find spurious optima. Left: An average example showing the formations learned on the two sets of data are very similar. Right: An unusual "bad" example showing more disagreement between the two data sets.

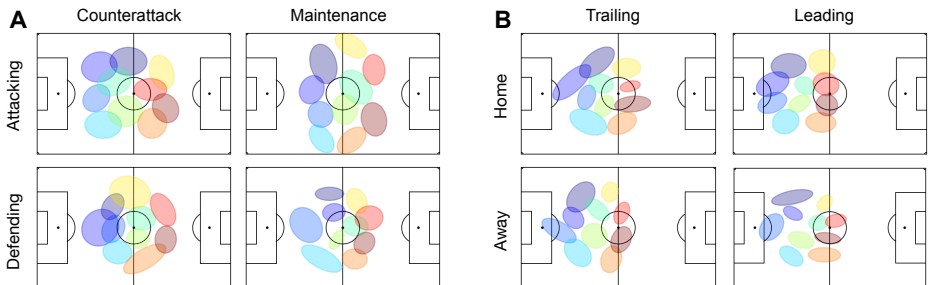

Figure 5: Context-specific templates. (A) We trained distinct templates of a given team while attacking and defending against certain modes (aka. "styles") of play. Data is aggregated over multiple games across the season. (B) We trained distinct templates of a given team while defending during the last 10 minutes of the games while trailing and leading, both home and away. Here we have added back in the average team (i.e. group) position to show the overall positioning on the pitch.

Figure 5 shows two such analyses this method unlocks [3]. On the left (A) we examine the formation of a team across an entire season when they are attacking in and defending against two very different "styles" of play (the very offensively aggressive counterattack, and a conservative "hold the ball" maintenance style) (Ruiz et al., 2017). Similarly, we can examine how a team positions itself when leading or trailing late in a game both at home or away (B). In addition to learning the formation, we can add back in the overall group positioning to see where on the pitch the team attempts to position itself. Additionally, we can learn and align the unique formations of every team across an entire season in a matter of minutes (see Figure 6 in A.6). Other potential analyses could include computing the formation after substitutions or analyzing how teams perform when certain individuals occupy a given role; such analyses are left to future work.

## 5 SUMMARY

For multi-agent systems with a high degree of structure such as that seen in team sport, we are able to learn a mapping which takes the set of agents to an ordered vector of agents without introducing undue entropy from permutation. In this work we have shown an improved method for learning the group representation of structured multi-agent data which is significantly faster. Additionally, the monotonically decreasing nature of its objective function provides stability. Our approach exploits the independence of the role-generating distributions during the template-learning phase and enforces the hard assignment of a single agent to a single role only during the final alignment step. This new approach, in combination with a smart choice of key-frame selection and initialization, allows for this representation to be learned over $n^3$ times faster- a factor of more than 1000 for a sport like soccer. By learning this representation, we are able to perform season-wide contextual and on-the-fly representation learning which were previously computationally prohibitive.

---

[3] A live, on-the-fly demonstration can be found at `https://www.dropbox.com/s/r33hphk5tubqvw5/alignment_onTheFlyApplications.mp4?dl=0`

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

# A APPENDIX

## A.1 SUMMARY OF THE FORMATION DISCOVERY ALGORITHM OF BIALKOWSKI ET AL. (2016)

To learn the per-role player distributions, the authors cast the problem as one of data paritioning where the goal is to minimize the overlap of an individual role $P_n(\boldsymbol{x})$ and that of the team $P(\boldsymbol{x})$ by placing a penalty $V_n$ on that overlap as defined by

$$V_n = -KL(P_n(\boldsymbol{x})\|P(\boldsymbol{x})) \tag{6}$$

where

$$KL(P(x)\|Q(x)) = \int P(x) \log \left(\frac{P(x)}{Q(x)}\right) dx. \tag{7}$$

Eq. 1 can then be written as

$$\boldsymbol{\mathcal{F}}^* = \arg\max_{\boldsymbol{\mathcal{F}}} V \tag{8}$$

Substituting Eq. 6 into Eq. 8 yields

$$\begin{aligned} V = &- \sum_{n=1}^{N} P(n) \int P(\boldsymbol{x}|n) \log P(\boldsymbol{x}|n) dx \\ &- \sum_{n=1}^{N} P(n) \int P(\boldsymbol{x}|n) \log P(\boldsymbol{x}) dx. \end{aligned} \tag{9}$$

Written in terms of the entropy $H(x) = -\int_{-\infty}^{+\infty} P(x) \log(P(x)) dx$ this simplifies to

$$V = -H(x) + \frac{1}{N} \sum_{n=1}^{N} H(\boldsymbol{x}|n) \tag{10}$$

yielding the final equation of $\boldsymbol{\mathcal{F}}^*$ as given in Eq. 3.

They approximate the solution by using expectation maximization (EM) as summarized here:

**Initialization**: The data is normalized such that the average team position in each frame is placed at the origin. The formation is initialized by assigning a player to a single role for the entire game. This allows the construction of $n$ independent distributions describing the $n$ different generating roles.

**E-Step**: An $n \times n$ cost matrix is computed for each frame which is based on the log-probability of each player being assigned to a particular role distribution.

**M-Step**: The Hungarian Algorithm (Kuhn, 1955) is used to (hard) assign each player to a specific role in each frame. Once all roles have been assigned for that iteration, the role distributions are recomputed.

**Termination**: The process repeats until convergence.

## A.2 RELATION TO GUMBEL-SINKHORN

In the original template discovery formulation, application of the Hungarian algorithm at each iteration made the strong requirement that a single player be assigned to a single cluster in each frame, that is, $\pi_{s,i} = \pi_{s,j} \forall i, j \in K, \forall s \in S$.

Our method treats each agent in each frame as fully independent; if we examined at the assignment probabilities for a given frame during training, the likelihood would exceed 1. Furthermore, during training, there are no restrictions on the values of each $\pi$ beyond the standard requirement that $\sum_{k=1}^{K} \pi_k = 1$. Thus at the end of the formation-learning step, there is no requirement that each

$\pi_k = \frac{1}{K}$. The requirement that each role be occupied by only a single agent in each frame and that weights are uniform is imposed only at the assignment step.

An approach based on Sinkhorn normalization can be seen as an intermediate between these two paradigms, allowing for *soft*-assignment of every agent in a frame to every cluster during the learning process.

Thus during training, a player could still contribute to multiple clusters, but the weighting of that contribution is restricted by Sinkhorn normalization which requires the assignment matrix in each frame to be doubly stochastic. Following the notation of Cruz et al. (2017), rows $R$ and columns $C$ are normalized according to

$$R_{i,j}(Q) = \frac{Q_{i,j}}{\sum_{k=1}^{l} Q_{i,k}}; \qquad C_{i,j}(Q) = \frac{Q_{i,j}}{\sum_{k=1}^{l} Q_{i,k}} \tag{11}$$

with the $n^{th}$ iteration defined recursively as

$$S^n(Q) = \begin{cases} Q & \text{if } n = 0 \\ C(R(S^{n-1}(Q))), & \text{otherwise.} \end{cases} \tag{12}$$

Similar to our approach, Sinkhorn-based methods employ hard assignment via the Hungarian algorithm only after training completes. The additional benefit of this approach is that the Sinkhorn normalization function is differentiable and thus amenable to gradient-based methods.

However as our approach is based on expectation maximization, the drawback to Sinkhorn-normalization is that the iterative normalization step is required in each frame at each EM iteration. Thus within the EM framework, it remains cost-prohibitive although gives final assignment results equivalent to both the original and current methods.

## A.3 OPTIMAL TEMPLATE LEARNING ALGORITHM

---

**Algorithm 1** Optimal Template Learning

---

**Input:**
    $U = \{U_1, U_2, \ldots, U_N\}$ unordered player positions
    $G^*$ a parent/global template

**Output:**
    $R = [R_1, R_2, \ldots, R_K]$ player positions ordered by role
    $\mathcal{F}^*$ the learned formation
    $\mathcal{T}^*$ the alignment template

---

**Normalization**

---

1:  normalize the positions in each frame so that the attacking team is going left to right
2:  center-normalize the positional data according to $M_{sn} = U_{sn} - \sum_{n=1}^{N} u_{sn} \quad \forall n \in N$
3:  format $M$ according to $f : \mathbb{R}^{S \times dN} \to \mathbb{R}^{SN \times d}$

---

**Formation Discovery**

---

4:  conduct K-Means clustering for initialization: K-Means($M, \mu_{init} = [\bar{M}_1, \bar{M}_2, \ldots, \bar{M}_N]$)
5:  **function** EIGENVALUERESETTINGGMM
6:     **while** lower bound average gain $<$ threshold **do**
7:        **if** $\frac{1}{r} < \frac{\lambda_{n1}}{\lambda_{n2}} < r \quad \forall n \in N$ **then**
8:           $\mu, \sigma, \pi \leftarrow$ GMM Update
9:        **else**
10:          $\mu, \sigma, \pi \leftarrow$ Soft K-Means Update
11:     **return** $\mathcal{F}^*$

---

**Template Alignment**

---

12: **function** ALIGNTEMPLATES($\mathcal{F}^*, G^*$)
13:     create cost matrix $C$ s.t. $C_{i,j}$ is the Mahalanobis distance between the $i^{th}$ distribution in $\mathcal{F}^*$ and the $j^{th}$ distribution in $G^*$
14:     apply Hungarian algorithm to find optimal assignment of $\mathcal{F}_i$ to $\mathcal{G}_j$
15:     **return** $\mathcal{T}^*$

---

**Role Assignment**

---

16: **function** APPLYALIGNMENT($R, \mathcal{T}^*$)
17:     **for** $s$ in $S$ **do**
18:        create cost matrix $C$ s.t. $C_{i,j}$ is the likelihood the $R_{s,1}$ ($i^{th}$-agent in frame $s$), belongs to the $j^{th}$ distribution of $\mathcal{T}^*$
19:        apply Hungarian algorithm to find optimal assignment of $R_{s,1}$ to $\mathcal{T}^*$
20:     **return** $R$

---

## A.4 TREE-BASED ALIGNMENT

This algorithm is adapted from Sha et al. (2017). The overall structure is the same with changes to the initializations captured in A.3.

---

**Algorithm 2** Learning process of tree-based alignment

---

**Input:** $R = \{U_1, U_2, \ldots, U_N\}$ unordered player positions
$\quad$ $G^*$ a parent/global template
$\quad$ $T = \emptyset, \mathcal{C} = \emptyset$
**Output:** $T, \mathcal{C}$
 1: **function** LEARNTREE($R$)
 2: $\quad$ **for** each layer $l$ **do**
 3: $\quad\quad$ **for** each node $n$ **do**
 4: $\quad\quad\quad$ learn $\mathcal{F}^{*l}_n$, $\mathcal{T}^{*l}_n$, $R^l_n$ from Algorithm A.3 with $G^* = \mathcal{T}^{*l-1}_n$ and $R = R^{l-1}_n$, the data contained in the parent node
 5: $\quad\quad\quad$ store $[\mathcal{T}^{*l}_1, ..., \mathcal{T}^{*l}_Z]$ in $T$
 6: $\quad\quad\quad$ compute reconstruction loss with Eq. 4
 7: $\quad\quad\quad$ terminate when stropping criterion is met
 8: $\quad\quad$ **for** each node $n$ **do**
 9: $\quad\quad\quad$ create $K$ initialization vectors by adding small amounts of noise $\epsilon$ to the cluster means of $\mathcal{T}^{*l}_n$
10: $\quad\quad\quad$ conduct K-Means on $R^l_n$ with different $K$
11: $\quad\quad\quad$ select cluster set $C^l_n$ that maximize $E$ partition $C^l_n$ to child nodes according to $C^l_n$ Store $[C^l_1, ..., C^l_Z]$ in $\mathcal{C}$
12: $\quad$ **return** $T, \mathcal{C}$

---

## A.5 RUN COMPLEXITY

Average per-iteration run-complexity for various methods.

Table 1: Algorithm complexity per iteration

| ALGORITHM | COMPLEXITY |
|---|---|
| Hungarian EM | $SN^5d^2$ |
| K-means | $SN^2d$ |
| GMM | $SN^2d^2$ |

## A.6 League-Wide Analysis

We ran our formation discovery and alignment algorithm on every team-game-period combination of an entire season. This analysis previously took upwards of 20 minutes to process a game under the original approach, but now can be run in less than 10 seconds a game.

Some teams consistently operate out of the same formation and therefore the templates and distribution centers are well isolated across the season (e.g. second row, far right). Others play different formations in different matches and therefore the game-to-game templates can vary dramatically (e.g. third row, second from the left; bottom row, second from the left). Our algorithm is able to learn these various templates and align them so that a common, structured representation can be used across matches.

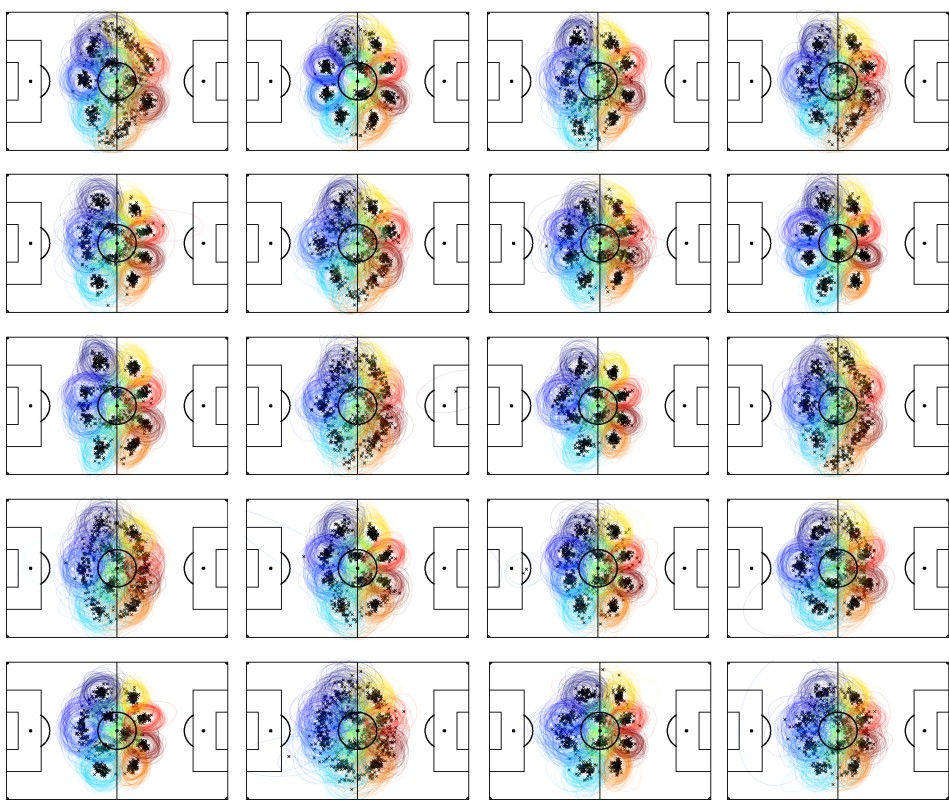

Figure 6: Discovered templates for each team across a season of professional soccer. The global template is learned by selecting data randomly across the season (all teams, all games). Each plot corresponds to a team and a template is learned for each half of every game and aligned to the global template. The centroids of each role-distribution are plotted in black.

