# OpenReview forum: "Improved Structural Discovery and Representation Learning of Multi-Agent Data"
_ICLR.cc/2020/Conference — Reject_

### Official Review · AnonReviewer1 · 2019-10-22
**Official Blind Review #1**

**Rating:** 6

**Review:**


  *Synopsis*:
  The paper proposes a new algorithm for template discovery and representation learning for unstructured multi-agent data with strong group roles (i.e. sports data). Their algorithm is several orders of magnitude faster because of a key assumption in the underlying structure of the position generating distributions of players conditioned on roles. They also formulate the problem in a novel way, splitting the structuring of the data into a three-step approach with each component requiring a different algorithm. This will allow the community to swap in new algorithms into the different components more simply than before. They finally provide some evidence of their algorithm working on a soccer dataset consisting of 380 games.

  Main Contributions:
  - A key insight into the underlying structure of the data, enabling an algorithm with orders of magnitude speed up.
  - A new framing of the problem.

  *Review*:

  The paper is well written, and seems to be a nice improvement over prior art in terms of runtime. I am not very familiar with the surrounding literature, but had a few concerns over the current paper that I feel should be addressed. Overall, I think this paper could be accepted unless other reviewers more familiar with the literature find critical flaws. I would, however, like to see Q2 addressed by the authors.

  Q1 It seems the key insight is that you are modeling the occupancy of space by an agent as a mixture of independent gaussian distributions. How reasonable is the assumption that each generating distribution conditioned on a role is independent of the others? Wouldn't a roles place on the field be dependent by the placement of other roles?

  Q2 How does the performance (i.e. accuracy) of your algorithm compare to prior approaches (i.e. the Sha or Bialkowski)? Or really any baseline? It is hard to judge this method in isolation (other than the impressive computational gains), especially for someone unfamiliar with this literature. Is there a way you could include a comparison in the final draft? If not, why?
      As a note: It seems you include some comparisons in section 4.3, 4.4, and 4.5, but it is unclear to me what the competitor is (i.e. A.3) and how it relates to the surrounding literature. If this is related to prior art, you should make it clear what algorithm this is in the main text, not relegating all discussion to the appendix.

  C1: It would be nice to include table 1 (appendix) in the main paper, or you could include a simplification of the runtime complexity of the K-Means and GMM algorithms in terms you use for the Hungarian algorithm in the main text. This would make it easer to see the significant gains your approach (i.e. using k-means) gets over prior approaches.

  *Minor Comments not taken into account in the review*
  - You should be more clear what the Hungarian algorithm is. Specifically, the Kuhn 1955 should be cited when the algorithm is first mentioned.
  - "global shape and structure the group" -> "global shape and structure of the group"
  - "'anchor' agent (Mehrasa et al.)" -> year for citation

----------
Post Review

I appreciate the author addressing my concerns and making clarifications here in discussion and in the text. I'm going to keep my score as a 6 primarily because I am not confident in the literature this paper is appealing to to truly advocate for its acceptance.



**Experience Assessment:**

I do not know much about this area.

**Review Assessment: Checking Correctness Of Derivations And Theory:**

I assessed the sensibility of the derivations and theory.

**Review Assessment: Checking Correctness Of Experiments:**

I assessed the sensibility of the experiments.

**Review Assessment: Thoroughness In Paper Reading:**

I read the paper at least twice and used my best judgement in assessing the paper.

---

> ### Author Response · Authors · 2019-11-07
> **Response to Review #1**
>
> Thank you for your questions and feedback.  We believe we can answer many of them here and make improvements to the text to address them.
>
> Q1- on the independence assumption:
> Yes, we do believe there is strong correlation among player positioning which is what the template clustering (steps 6 and 7) aim to capture.  However, we cannot capture that correlation (through a 20dimensional distribution, for example) until an ordering is determined.  A key point here is that we have broken the process into a template learning step and an assignment step.  During the template learning step, we seek to find the distribution (which we model as a mixture of independent Guassians) which maximizes the likelihood of the data.  Then, during assignment, we enforce the requirement that each role must be occupied by a single player- this introduces some dependence between the players and alters the observed role distributions slightly as seen in to bottom plot (step 6) of Figure 2.  Once an ordering has been set, we can then capture the full dependencies between the players by fitting a 20-dim distribution (or mixture of 20-dim distributions) to the data.  This gives us our template clusters (step 7).  Following the tree-based method of Sha et al., these steps then get repeated: for each cluster we first treat the players as independent handle the permutation issue and allow us to refine the template, players are assigned to new refined roles in the new subtemplates, and then a 20-dim distribution is fit using those new roles.
>
> Q2- On Performance (accuracy):
> We would kindly ask the reviewer to further explain what types of comparison or metrics they would find useful.  As all of these methods discussed here are unsupervised, therefore there is no notion of "accuracy" for a template.  Similarly there is no accuracy measure for a representation.  Efficiency (i.e. complexity) is a measure we discuss in section 4.2.
>
> We would be able to compare the distributions found by the two approaches.  Focusing on Bialkowski for simplicity (Sha follows the same general approach just in an iterative fashion to generate sub-templates from the template), they also seek to find the "template" F* that maximizes the likelihood of the data using EM.  Consider Figure 2.  Our approach first solves for the optimal formation F* that maximizes the likelihood of the data.  It then performs assignment to enforce one player per role.  Whatever this new distribution is it must explain the data no better than F*, by definition.  The difference in these two distributions is seen in Step 6 of Figure 2. In contrast, Bialkowski directly solves for this final distribution (the textured ellipses of this figure).  So their final ordering and role-based distributions should be effectively identical to ours (sans convergence issues), but their "F*" is not as optimal as it could be because it has already enforced this additional assignment constraint.
>
> Would quantifying the average difference (i.e. Bhattacharyya distance) between these distributions (our F* and the final observed role-based distributions) be useful?  This would give a "best-case" difference (not "improvement", but smallest difference, since it is unsupervised) between our method and Bialkowski. This is a key point we struggled with communicating and we will attempt to improve the text to make this clearer as all of the reviewers have asked something similar.
>
> The algorithm in A.3 is the current method, not the original algorithm of Bialkowski.  The method is explained graphically at a high-level and with discussion in the main text with the specific implementation details in the appendix for those interested in that formulation.
>
>
> On table 1:
> Table 1 was originally in the main paper, but due to the single-column formatting and large spatial padding of the this style set, it looked very empty and awkward on the page (that small table takes up about 1/4 of a page with this styling).  We will include the other runtime information in the text as the reviewer suggests.
>
>
> On the minor comments:
> Thank you for these finds- we will make the necessary corrections. We thank the reviewer for catching the missed reference- we had this reference originally and it is in the bibliography, but the original reference in the main text must have been cut during editing- we will add this back in at the first usage in the current text.  We will correct the missed date in the second reference as well.

---

> > ### Comment · AnonReviewer1 · 2019-11-13
> > **Thank you for the response**
> >
> > Thank you for your response, and effort in making the main contribution of your paper more clear.
> >
> > As far as your question. I am unsure, as I am quite unfamiliar with this literature. Although it seems like Reviewer 1 would have a better idea on what comparisons are appropriate.
> >
> > As for your clarifications in the text. Could you highlight/point to the differences?

---

> > > ### Author Response · Authors · 2019-11-15
> > > **Performance Measures and Text Improvements**
> > >
> > > We have added a new section (4.3) which compares the likelihood of the data under the formations learned via the two primary methods (the earlier hard-assignment method and our proposed approach) as well as added the comparison hard-assignment approach to the compression analysis (4.4).  We also made a correction to what is now Figure 3B(right) which had not been included properly.  We believe these analyses will address the reviewers request for comparisons and metrics.
> > >
> > > There have been text modifications throughout the text including- we believe pronoun references were made clearer and more explicit and added sentences which we hope add clarity.  In particular, we have tried to emphasize in several places (especially earlier in the text) that both the hard-assignment and current method are attempting to solve the same optimization problem, therefore attempting to converge to the same solution.  Additionally, the newly included results we believe will help make the similarities and differences clearer.

---

### Official Review · AnonReviewer2 · 2019-10-22
**Official Blind Review #2**

**Rating:** 3

**Review:**

In this paper authors describe a way to quickly model interaction between many entities, in the form of mixture of Gaussians. They provide huge computational complexity improvements over Bialkowski et al. work, which allows them to analyse big datasets.

Paper, in its current form, lacks quantitative evaluation, comparison to other methods (heavily cited in the paper), and any well established applications that can be quantified. Given that the method described is a composition of many known techniques, and some new tricks, it is hard to assess the value of these contributions without seeing more unified evaluation methodology. All the provided results (apart from computational complexity wrt. specific approach, which are quite amazing in isolation!) are qualitative and descriptive in nature, which together with the fact of being applied to a single, specific dataset, with makes it hard to judge any generality. This is a common problem with various "structure discovery methods" such as clustering techniques, but an important one to address.

It is also unclear to reviewer, whether the only (main?) claim authors want to make is that they provide speed-up to the existing solution. if this was the case, it is really hard to see how each of the proposed steps guarantees the same outputs as previous approaches, while at the same time only affecting speed. In particular authors make the claim of improvement stability, yet do not provide empirical evaluation showing these benefits over previous methods.

Please compare these to results reported e.g. in the cited work of Yeh et al. "Diverse Generation for Multi-agent Sports Games".

Overall, in reviewer's opinion, results looks quite incremental, and are heavily concentrated around specific paper of Bialkowski et al., and as such might see limited interest in the ICLR community.

Minor comments:
- please unify use of quotations (authors use closing quotations for both opening and closing quotes)
- is there an extra | symbol in eq. (4)?


Please note, that reviewer has little experience in this particular branch of the field, and thus some of the issues mentioned could come from misunderstanding the standards.

**Experience Assessment:**

I do not know much about this area.

**Review Assessment: Checking Correctness Of Derivations And Theory:**

I assessed the sensibility of the derivations and theory.

**Review Assessment: Checking Correctness Of Experiments:**

I assessed the sensibility of the experiments.

**Review Assessment: Thoroughness In Paper Reading:**

I read the paper at least twice and used my best judgement in assessing the paper.

---

> ### Author Response · Authors · 2019-11-07
> **Response to Review #2**
>
> Thank you for your comments and questions.  We believe we are able to address your concerns in the response below and with minor modifications to the work.
>
>
> On the main contribution of the work:
> Performance is largely the main "benefit" of this approach.  Besides significantly reducing computation time (which will be tremendously valuable for the other researchers who use and baseline against this approach), this allows for certain interesting domain-specific applications (several given in Section 4.5) that tend to be more focused on the "live" formation discovery, which tend to be more interesting to the sports community.
>
> We believe the main contribution is in clarifying the methodology and identifying the true F*.   Bialkowski seeks to find a template F* that maximize the likelihood of the data.  However, because hard assignment is applied during training, F* is never actually discovered.  Our approach allows us to first discover F*, and then to apply role-assignment to get the optimal ordering.  This has important practical implications (i.e. speed), application implications (i.e. a better formation,  "live" sports analysis), but also key theoretical implications (i.e. our approach actually finds F* and avoids spurious solutions because our M-step is truly maximal).
>
> The improved stability is theoretically guaranteed because our M-step is truly maximal.  As far as an empirical comparison on stability, would the reviewer find it useful to show a few examples of failure cases under the previous method?
>
> Regarding comparison to other methods, please see the response to Question 2 of Reviewer 3.  We believe that will address those concerns.
>
> Regarding applications, please see the response to Reviewer 1.  Our approach allows for all of the applications that have previously existed to still be performed, now just much faster.  The "new" applications are mostly focused around different types of analysis that can be performed "live" as a result of creating very specific templates under highly specific contexts (two examples are given in section 4.5).  We felt these types of analyses would be more appropriate to a sports-analytics community and less relevant to the broader ML community of ICLR.  However, since all of the reviewers have asked about these types of analysis, we can provide additional examples in the appendix.  We will also improve the text to further emphasize this point.
> ​
>
> On comparison to Yeh et al.:
> Yeh et al. uses a graph-neural network which is permutation-equivariant as discussed in Section 2.2.  Thus they do not have a template or equivalent representation to compare to.   In their work, they do an extensive comparison of their approach using a GNN with ordered data to non-graph-based deep learning approaches which rely on role or tree-based alignment for the underlying representation learning.  Their goal is to show that their GNN with no ordering performs better than "traditional" NN with ordering for trajectory generation.  Their method does not have an equivalent representation to compare to ours directly.  Our focus here is not on the task of trajectory generation but on representation learning.
>
> That being said, our approach can be used to generate effectively identical representations to what they call the "template" and "tree" based representation, just faster (please see the response to Q2 of the Review 1 for a more detailed explanation as to why this is true).  With effectively identical representation, the secondary task (i.e. trajectory generation) would also produce essentially identical results to the template and tree methods that they list in their comparisons.
>
>
> On the contribution to the ICLR community:
> Please see the comments for Reviewer 1 which discusses the key work being done in this field and has relied on the former approach in the past.  Given the visibility and extent to which the former approach was used, we believe it is important to share with the broader ML community which is using that approach this important improvement.
>
> On the minor comments:
> We thank the reviewer for the corrections and will make these improvements.

---

> > ### Author Response · Authors · 2019-11-15
> > **Evaluation**
> >
> > We have added a new section (4.3) which includes several metrics comparing the formations obtained via hard-assignment and the new method.  Additionally, we now include the hard-assignment in the compression analysis of section 4.4.  We believe these additions will address the reviewers concerns.  We thank the reviewer for the request to add these metrics because we believe they greatly enhance the work.  Hopefully in addition to providing the desired quantitative comparisons, they also highlight the fact that the hard assignment and new method are both solving for the same objective function and therefore approach  (quantifiably) extremely similar solutions.

---

### Official Review · AnonReviewer3 · 2019-10-25
**Official Blind Review #3**

**Rating:** 1

**Review:**

After author rebuttal:
Thank you for your response to the review. I appreciate your point that sport applications can be interesting to this community, and your paper makes an important contribution in that direction. With that said, for a paper to be appealing to the ICLR audience, there should be more clear novelty in the method, and stronger experimental evaluation showing novel insights or out-performance over previous baselines.


==================================================================
The paper addresses a problem in the context of understanding formations of sport player in a play field. The problem is that players form an ומ-ordered sets of entities and to analyze them, one would like to keep some canonical representation that allows us to compare formations across time. The idea of this paper is to assign roles to each players (name each member of the set) based on the distribution of their spatial locations on a field. Once this representation is learned, future work may use it for downstream problems, which are not discussed in this paper.

General:

The paper is very clearly written, the method makes sense and the related work are both well explained.
I have two main concerns: First, the scope of the paper may be too narrow for a conference like ICLR, as the method part is fundamentally a clustering method, and the application is very specific. Second, I was expecting that the paper shows interesting results that can be obtained once the representation is learned, but this are lacking from the paper.
Based on these two concerns, I dont think the paper would be a good fit for ICLR.

To make the paper more appealing to the ML community, my recommendation to the authors is to  show what can be achieved with the new representation, that was impossible with previous representations. For example, it is claimed that faster learning allows to compute it season wide and on-the-fly. What are useful tasks that your rep' can help with? How do these tasks perform with the new rep?

Other comments and questions.

-- What about other competing representations. Would it be useful to look at movement trajectories instead of distribution of isolated locations (cluster lines instead of points)?
-- What happens when a player is replaced by another player of the same role. Wouldnt its distribution change? Dont you want your rep' to reflect such events?


**Experience Assessment:**

I have read many papers in this area.

**Review Assessment: Checking Correctness Of Derivations And Theory:**

N/A

**Review Assessment: Checking Correctness Of Experiments:**

I assessed the sensibility of the experiments.

**Review Assessment: Thoroughness In Paper Reading:**

N/A

---

> ### Author Response · Authors · 2019-11-07
> **Response to Review #3**
>
> We thank the reviewer for their response and feedback and believe we can address the concerns that were raised.
>
>
> On the concern about the placement at ICLR:
> The machine learning community has increasingly embraced the domain of sport as an important and interesting domain in which to study multi-agent motion and interaction.  In recent years, there have been a number of papers accepted to top conferences which have used or baselined against the previous approach our current work improves upon.  Several of these papers include Zheng et al. (NeurIPS 2016), Le et al. (ICLM 2017),  Felsen et al. (ECCV 2018), Zhan et al. (ICLR 2019),  Yeh et al. (CVPR 2019), among others.
>
> As a result, we felt it was of significant importance to the community to publish in a venue that these researchers are engaged in so the field would be informed of this important improvement.  By utilizing the new method, these researchers could save tremendous time in their work- both in compute and in the time required to implement the algorithm.
>
> It is true that we get some sport-specific analyses "for free" as a result of this approach, but we feel the main contribution of this work is providing the community with a faster, more reliable means of obtaining a useful representation for studying multi-agent motion in sport.  We believe this will be of particular importance to the RL community with the recent release of Google's research football RL environment (Kruach 2019), which was announced at ICML in 2019.  We believe that ICLR is the best venue for reaching researchers with this important change to a standard method in the field.
>
>
> On the concern around new applications:
> Drawing on the references above, there has been tremendous work done by the machine learning community using alignment-based approaches in sport. One of the main positives of this new approach is that it does not invalidate any of those works, only reduces the preprocessing time.  Anything that could be done under the previous approach (trajectory search/retrieval, autoencoding, trajectory prediction/generation, passing analysis, metric creation, formation prediction, etc.), is still feasible. To introduce a brand new application, does not speak to the benefits of the representation learning aspect of this work other than to say it is faster (which can be said without introducing a new task).  For that reason we felt introducing a specific secondary task diluted the story- all secondary tasks are still valid. The core representation is the same under the old and new approaches, this is now solved for in a much faster manner, has some stability gains, and nice theoretical properties (please see the Response to Q2 of Review #1).
>
> "New" applications which are gained are ones which were previously theoretically feasible but cost-prohibitive like the context-specific analysis which we showed ("on-the-fly" analysis in Figure 5, season-wide analysis in Figure 6).  There are other similar "drill-down" analyses that could be done, but we felt these became too sport-specific focused and took away from the important theoretical contribution here.
>
>
> On the question of competing representations:
> Yes, it is entirely possible to cluster player trajectories instead of single frame locations.  This was done in Sha et al. 2017 and used by Felsen et al. 2018.  The beauty of the approach introduced here is that it can be applied on trajectories as well as single frames and eliminates the same computational bottleneck step.  Felsen et al. 2018 and Yeh et al. 2019 have very thorough analyses on the impact of representation on trajectory generation.  Our method could be used to produce essentially identical results to those obtained with the frame-base role and tree-based trajectory alignment.
>
>
> On the question of substitutions:
> The goal of role assignment is to capture the role a player is in, independent of identity.  Eliminating that identity dependence is the very goal of this process.  Because players switch positions and substitute for one another, identity-based representation is much less compressive than role-based representation (Figure 4).
>
> That being said, in football, for example, formations often change after substitutions for tactical reasons.  Our approach allows us to learn a new template after every substitution if desired and compare them; it is no longer time-prohibitive as it was under the old approach.  We would generally classify this under "context specific formations" analysis (section 4.5), but can make this particular use-case more explicit in the text.  Once these multiple templates are generated,  a wide-variety of analyses are feasible; again we felt this was another example of the same "type" of analysis shown in Figure 5 and that particular analysis would be more relevant in a sport-focused publication but could include an example if the reviewer feels it would be illuminating to the reader here.

---

> > ### Author Response · Authors · 2019-11-15
> > **Application- video**
> >
> > We have included a video which we believe will show some of the analyses that can be performed with the ability to compute the formation on-the-fly.  To maintain the anonymity of the double-blind review process, we have made a simple video of an application.  Once the review process has ended, we can replace this video with a one that does not have to mask out identifying information, but would contain the same information.  Note that near the end of the video, we examine how the formation changes before/after player substitutions as the reviewer asked about.

---

### Decision · Program_Chairs · 2019-12-19

**Decision:**

Reject

**Comment:**

The work addresses the problem of inferring group structure from unstructured data in multi-agent learning settings, proposing a novel approach that has key computational / run time advantages over a prior approach. A key limitation raised by reviewers is the limited quantitative evaluation and comparison to previous approaches, as well as a resulting set of general insights into advantages of the proposed approach compared to prior work (beyond computational benefits). While some of the key limitations were addressed in the rebuttal, the contribution in its current form remains too narrow. The paper is not ready for publication at ICLR at this stage.